# Combining Genetic Gain and Diversity in Plant Breeding: Heritability of Root Selection in Wheat Populations

Johannes Timaeus *[ID], Odette Denise Weedon [ID] and Maria Renate Finckh [ID]

Department of Ecological Plant Protection, University of Kassel, Nordbahnhofstr. 1 a,
37213 Witzenhausen, Germany; odetteweedon@uni-kassel.de (O.D.W.); mfinckh@uni-kassel.de (M.R.F.)
* Correspondence: johannes.timaeus@uni-kassel.de

**Abstract:** To increase the resilience of agroecological farming systems against weeds, pests, and pathogens, evolutionary breeding of diversified crop populations is highly promising. A fundamental challenge in population breeding is to combine effective selection and breeding progress while maintaining intraspecific diversity. A hydroponic system was tested for its suitability to non-destructively select root traits on a population level in order to achieve genetic gain and maintain diversity. Forty wheat progenies were selected for long seminal root length (SRL) and 40 for short SRL from a wheat composite cross population grown in a hydroponic system. Wheat progenies were multiplied, and a subset evaluated again in a hydroponic system. Preliminary tests in soil and competition experiments with a model weed were performed. The hydroponic selection for long SRL led to an increase of SRL by 1.6 cm (11.6%) in a single generation. Heritability for selection of SRL was 0.59. Selecting for short SRL had no effect. The preliminary soil-based test confirmed increased shoot length but not increased SRL. Preliminary competition experiments point to slightly improved competitive response of wheat progenies but no improved competitive effect on mustard. These results indicate a heritable selection effect for SRL on a population level, combining genetic gain and intraspecific diversity.

**Keywords:** composite cross population; organic plant breeding; crop diversity; genetic gain; early vigor

## 1. Introduction

There is a strong need for cultivars that are bred and optimized for agroecological and organic farming systems [1]. An increasing organic sector requires cultivars adapted to organic farming [2]. Reduced diversity in cropping systems is a major cause of their vulnerability to weeds, pests, and pathogens, requiring intense measures of plant protection either chemically or mechanically [3]. In addition, homogeneous cropping systems can also be vulnerable to abiotic stresses such as increased weather variability under climate changes found in a recent study for wheat cultivars [4]. Evolutionary breeding of composite cross populations (CCPs) has been developed to foster intraspecific diversity to enhance resilience against biotic and abiotic stresses that occur in organic and agroecologically managed farming systems [5–7]. Recent long-term studies provide robust evidence, by analyzing yield data for up to 13 years with stability indices, that CCPs are highly adaptive to environmental stress and provide higher yield stability than genetically homogeneous line cultivars in wheat [8–10]. Diversity remains high in wheat populations even after many years if sufficient effective population sizes are maintained [11]. Seeds of commercial wheat populations are already available [12]. Recently, a CCP evolved at the University of Kassel under organic conditions for 13 years was licensed under an open source agreement under the name of 'EQuality' [13]. The new EU Organic Regulation 2018/848 will come into force in 2022 and will provide a legal framework for heterogeneous organic materials, enabling their further mainstreaming.

A fundamental challenge in population breeding is to combine effective selection methods and breeding progress for key traits while maintaining intraspecific diversity

that confers stress resilience. One strategy is to select lines that vary in a target trait while being homogeneous for traits influencing agronomy such as the time of ripening, the so-called multiline approach [14]. This approach is very resource-demanding due to multiple backcrosses that are required. Another less resource-demanding approach is to carry out non-destructive mass selection of CCPs that maintains population level diversity.

Roots and their early development play a pivotal role in the efficient uptake of nutrients [15,16] and interaction with the soil microbiome [17–19]. Soil-less screening methods such as hydroponic systems allow for the efficient phenotyping of root traits in a non-destructive manner [20,21]. A hydroponic system was successfully used to investigate changes in wheat CCPs root development after six and ten years of evolution in different farming systems [22,23]. In wheat CCPs evolving under organic farming conditions at the University of Kassel, seminal root length and root weight increased and total root length and specific root length decreased compared to CCPs evolving in a conventional system. Vijaya Bhaskar et al. [24] showed that seminal root and shoot length of two-week-old plants in hydroponics was a good proxy for more complex plant traits such as total root length and early soil cover in the field. Seminal root length can easily be assessed in hydroponics in a non-destructive manner. This suggests that, provided enough individuals are selected to conserve population diversity, direct non-destructive selection for seminal root length in hydroponic conditions should speed up and complement evolutionary breeding.

In this study, we tested whether a hydroponic system can be used to non-destructively select root traits from the wheat CCPs for high early vigor in order to increase genetic gain, while maintaining population diversity. Thus far, studies in the field of wheat breeding focus either strongly on genetic gain [25] or maintaining diversity [11]. This study aims to combine both aspects in a novel way, enabling the selection of new genetic wheat material that is genetically diverse and carries highly valuable root traits, while addressing the following research questions:

1.  Is selection for seminal root length in hydroponics heritable?
2.  How strong is the genetic gain achieved in one breeding cycle on a population level?

To answer these questions, wheat plants were selected from a CCP for seminal root length in hydroponics and grown to maturity in soil substrate. Progenies of these plants were evaluated for selection effects in a hydroponic system. In addition, progenies with sufficient seed quantities available were additionally tested in soil substrate. Finally, the parental CCP and selected progenies were evaluated for their competitive effect on and competitive response to the model weed mustard.

## 2. Materials and Methods

### 2.1. Plant Material

The wheat composite cross population YQ used in this study was created in 2001 by the Organic Research Centre and the John Innes Institute in the UK. Out of a half-diallel cross of twenty wheat varieties, the progenies of the crosses of eight high-yielding (Y) x 11 baking-quality (Q) parents, plus all 19 parents crossed with cultivar Bezostaya, were combined [26]. Since 2005 (F5), the YQ CCP has been maintained at the University of Kassel under organic conditions as OYQ CCP without conscious selection, apart from removal of plants taller than 130 cm in the first three years. In 2006/07 (F6), it was split into two parallel non-mixing populations (OYQI and OYQII). Parallel populations allow for distinguishing random changes due to genetic drift and changes due to selective effects of the natural environment or farming systems [10]. The OYQII F16 harvested in 2017 was used in this study. In addition, four contrasting commercial wheat cultivars were included as reference. Elixer is a short high-yielding cultivar, and Capo is a tall baking-quality cultivar. Kolompos is an early and fast-growing Hungarian baking-quality cultivar with high early vigor and early heading [27]. In addition to these conventionally bred cultivars, Butaro, an organically bred tall baking-quality cultivar, was included.

## 2.2. Hydroponic System

For the selection experiment, a hydroponic system based on Bertholdsson et al. [22] and described in detail in Vijaya Bhaskar et al. [24] was used. A complete nutrient solution with a phosphate buffer (pH 6.5) [20] was used. Stock solutions of macronutrients (for 2 L: $Na_2HPO_42H_2O$—39 g, $KH2PO4$—68 g, $KCL$—37 g, $MgSO_47H_2O$—61 g, $Ca(NO_3)_2$—118 g) and micronutrients (for 1 L: $FeCl_36H_2O$—27.53 g, $MnCl_24H2O$—1.39 g, $ZnSO_47H_2O$—0.86 g, $CuSO_45H_2O$—0.20 g; $H_3BO_3$—0.099 g and $Na_2MoO_42H_2O$—3.40 g) were prepared separately. Per 20 L container, 200 mL of the macronutrient and 100 mL of the micronutrient stock solution were added and filled with deionized water to achieve 2 mM N concentration. The nutrient solution was renewed after seven and ten days and aerated continuously by pumping air through the solution with a TetraTec APS (5l/min). The containers were placed in a greenhouse with 18/12 °C (day/night) temperature regime for 14 days at a photoperiod of 16 h day/8 h night. Seeds were placed in strips of cardboard with the embryo facing downwards. The lower part of the cardboard was ironed to prevent seeds from falling down, and filter paper strips were used as wicks and suspended over special frames into the hydroponic containers. Per container, 14 rows with 12 seeds per row were accommodated. Row 1 and 14 as well as the outer plants of Rows 2 to 13 were sown with Capo as edge plants.

## 2.3. Hydroponic Root Length Selection, Seed Propagation, and Evaluation

On 4th November 2018, four containers were sown with OYQII F16 seeds. After 14 days, the 40 individuals with the shortest seminal roots (Sel_S) and 40 individuals with the longest seminal roots (Sel_L) were selected out of 401 undamaged well-growing individuals. All 80 selected plants were immediately transferred into commercial peat-based potting soil without fertilizer in quick pots and vernalized for six weeks at 5 °C with artificial lighting (photoperiod of 16 h day/8 h night) and then transplanted into 10 L pots. The pots contained a mixture of 30% C-horizon soil, 60% commercial potting soil, 10% sand, and 20 g of hair meal pellets (14% N). Plants were grown for seed propagation in an unheated greenhouse without artificial lightning. Seeds of 78 plants could be harvested in June 2019, and thousand grain weight (TGW) and number of seeds produced were determined. Nine Sel_L and nine Sel_S progenies that produced sufficient seed for simultaneous evaluation in hydroponics and in pot tests while leaving enough seed for further propagation were evaluated in hydroponics in January 2020 with six replicates. Each half row per container was randomly assigned to a progeny or reference variety with six seeds each, resulting in a randomized complete block design. Replicates were sown consecutively every three days to allow sufficient time for handling and processing the plant samples after each harvest. Seminal root length (SRL) and primary shoot length (SL) were measured with a ruler. Fresh weights of roots (RFW) and shoots (SFW) were measured directly after harvest and dry weights (RDW and SDW) determined after drying at 105 °C for 24 h.

In a first preliminary study, the same selected progenies evaluated in hydroponics were grown in root training pots of 20 cm height to allow enough space for root growth. The soil substrate consisted of 40% C-horizon soil, 24% compost (1.1% N), 20% peat, and 16% sand. Biodegradable root sleeves were placed in the pots and then filled with the soil substrate. Two replicates with five plants per genotype were sown with genotypes randomized within each block. A single seed was sown per pot. The limited number of seeds available also constrained the number of replicates in this study. One row of Capo was sown in border pots to avoid edge effects. The plants were grown under the same conditions as the hydroponic experiment for four weeks. SRL, SL, RFW, RDW, SFW, and SDW were measured as in the hydroponic system after the careful cleaning of the roots.

## 2.4. Competition Experiment

Of the eighteen progenies used in hydroponics, three Sel_S (213, 46, 37) and three Sel_L (145, 191, 56) progenies were selected randomly for this experiment together with the parent OYQII CCP, Kolompos and Elixer. White mustard (*Sinapis alba*) was used as a model

weed. Since white mustard is a domesticated plant, this model weed not only ensures homogeneous germination but also shares traits with relevant weeds in agriculture, such as wild mustard (*Sinapis arvensis*). The competition experiment was carried out with the same substrate as described above. Pot size was $60 \times 17.5 \times 14.5$ cm, and 18.4 g (14% N) of hair meal pellets were added to each pot. The experiment included five replicates, each containing all genotypes randomly assigned to pots. Each pot was subdivided into two sections with a plastic barrier. In one section, two rows of seven wheat seeds were sown, while in the second section, two rows of wheat were sown, in between which one row of seven mustard seeds was sown. Sowing density was equivalent to standard sowing density for wheat in organic farming, i.e., 350 seeds/m$^2$. Single mustard rows without wheat were included as reference. Plants were grown for six weeks. After carefully cleaning the roots, RFW, RDW, SFW, SDW, and SL were determined. In addition, leaf length and width for the youngest fully developed leaf and individual plant length were measured with a ruler. Leaf area (LA) was estimated by following Chanda and Singh [28] as:

$$LA = w \times le \times 0.75 \tag{1}$$

where $w$ is leaf width and le leaf length. Plant cover was estimated visually (%).

### 2.5. Data Handling and Statistical Anlysis

All statistics were calculated in R [29]. Dplyr [30] was used for data aggregation and handling, ggplot2 [31] and ggpubr [32] for plotting. Normality was assessed by histograms and QQ-plots and variance heteroscedasticity was tested by Levene's Test for model residuals and residual graphing methods. Linear mixed effects models were constructed with the lme4 package [33]. Observed heteroscedasticity was accounted for by variance weighing in the nlme package [34]. Estimated marginal means were calculated with the emmeans package [35] followed by a post hoc test with pairwise comparison and Holm correction. Random effects were included to account for nested and non-random experimental structure such as replicate blocks or genotypes grown in rows/pots in each replicate. Where appropriate, models were made either for individual selected progenies or for all Sel_L and Sel_S progenies combined as genotype groups. For the hydroponic evaluation experiment, traits were measured for individual plants, and we included replicates as fixed effects and plant rows as random effects nested into the replicate for the statistical model:

$$response \sim genotype + replicate + 1|replicate : row \tag{2}$$

For the soil evaluation experiment, the model was:

$$response \sim genotype + 1|replicate \tag{3}$$

For the competition experiment, the model for wheat traits was:

$$response \sim genotype \text{ x } system + replicate + 1|replicate : pot \tag{4}$$

The model for mustard in the competition experiment was:

$$response \sim genotype + 1|replicate \tag{5}$$

To test differences in TGW for the different wheat genotypes, a Kruskal–Wallis test was applied. A full list of all models is given in the Supplementary Materials (Table S1), also documenting the use of lmes or nlmes based on the observed variance structure. Competitive interactions were divided into competitive effect of the crop plant on the weed plant (weed suppression) and competitive response of crop plant to the weed (weed

tolerance) following Wang et al. [36]. We calculated competitive response (*Cr*) of wheat as relative change of wheat in mixture compared to wheat in monoculture calculated as:

$$Cr = \frac{T_{mix}}{T_{mono}} - 1 * 100 \tag{6}$$

where *T* is the respective trait in mixtures and monoculture. Spearman correlation coefficients were calculated with the GGally package [37]. SRL, SL, RDW, and SDW were averaged for each genotype across the hydroponic experiment for correlation with TGW as this was determined from the seed bags and not for each replicate. For a more precise estimation, an additional correlation with individual plants was calculated between SL, SRL, RDW, and SDW. Heritability (h) for SRL was calculated following Becker [38]:

$$h = \sqrt{\frac{R}{S}} \tag{7}$$

where *S* is strength of selection, i.e., the difference of the mean of the trait of the parental population ($T_p$) and the selected subpopulation ($T_s$):

$$S = T_s - T_p \tag{8}$$

*R* is the breeding progress or genetic gain of a trait as difference of the mean of the parental population and the *F*1 of the selected subpopulation.

$$R = T_{sF1} - T_p \tag{9}$$

## 3. Results

### 3.1. Hydroponic Selection for Root Phenotype and Soil Test

TGW of the OYQII CCP, Sel_L, and Sel_S progenies did not differ significantly (Figure 1). Kolompos had the highest TGW (49 ± 1 g) and Capo the lowest (40 ± 1 g). Across all genotypes, there was no significant correlation between TGW and SL or SRL (both rs = −0.05). The correlations between TGW and RDW (rs = 0.62, *p* < 0.01) and SDW (rs = 0.43, *p* < 0.05) were moderate (Figure S1). SRL and SL were moderately correlated with each other for OYQII (rs = 0.67, *p* < 0.001) and for Sel_L (rs = 0.55, *p* < 0.001) but considerably less for Sel_S (rs = 0.30, *p* < 0.001) (Figure S2).

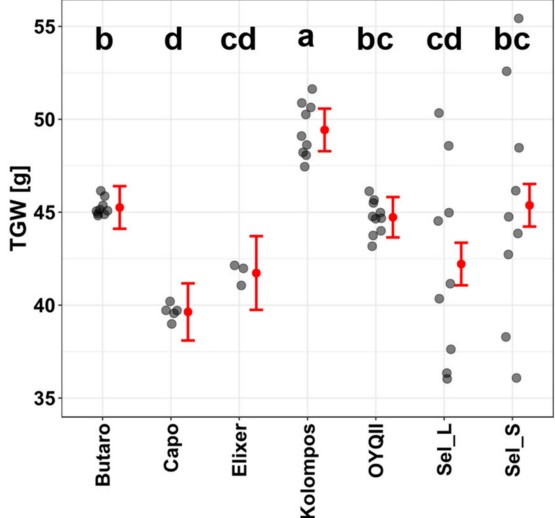

**Figure 1.** Thousand grain weight (TGW) of the materials used for the experiments. Raw data are plotted (black dots) and estimated marginal means with standard error. Different letters indicate significant differences at *p* < 0.05 based on a Kruskal–Wallis test.

Elixer had the shortest seminal roots (12.0 cm) and Kolompos the longest (16.3 cm). A significant population-level shift towards increased SRL from OYQII (13.8 cm) was achieved in the Sel_L progenies (15.4 cm, $p < 0.05$), but no significant changes became apparent in the Sel_S (13.6 cm) progenies (Figure 2A,C). Increases in SRL compared to OYQII were statistically significant in four of the nine Sel_L progenies while SRL of only one Sel_S line was significantly shorter (Table 1, Figure S3). On average, the selection effect for increased SRL was thus 1.6 cm in a single generation, i.e., an increase of 11.6%. Heritability based on means for SRL for Sel_L lines was 0.59 and for Sel_S lines was 0.21.

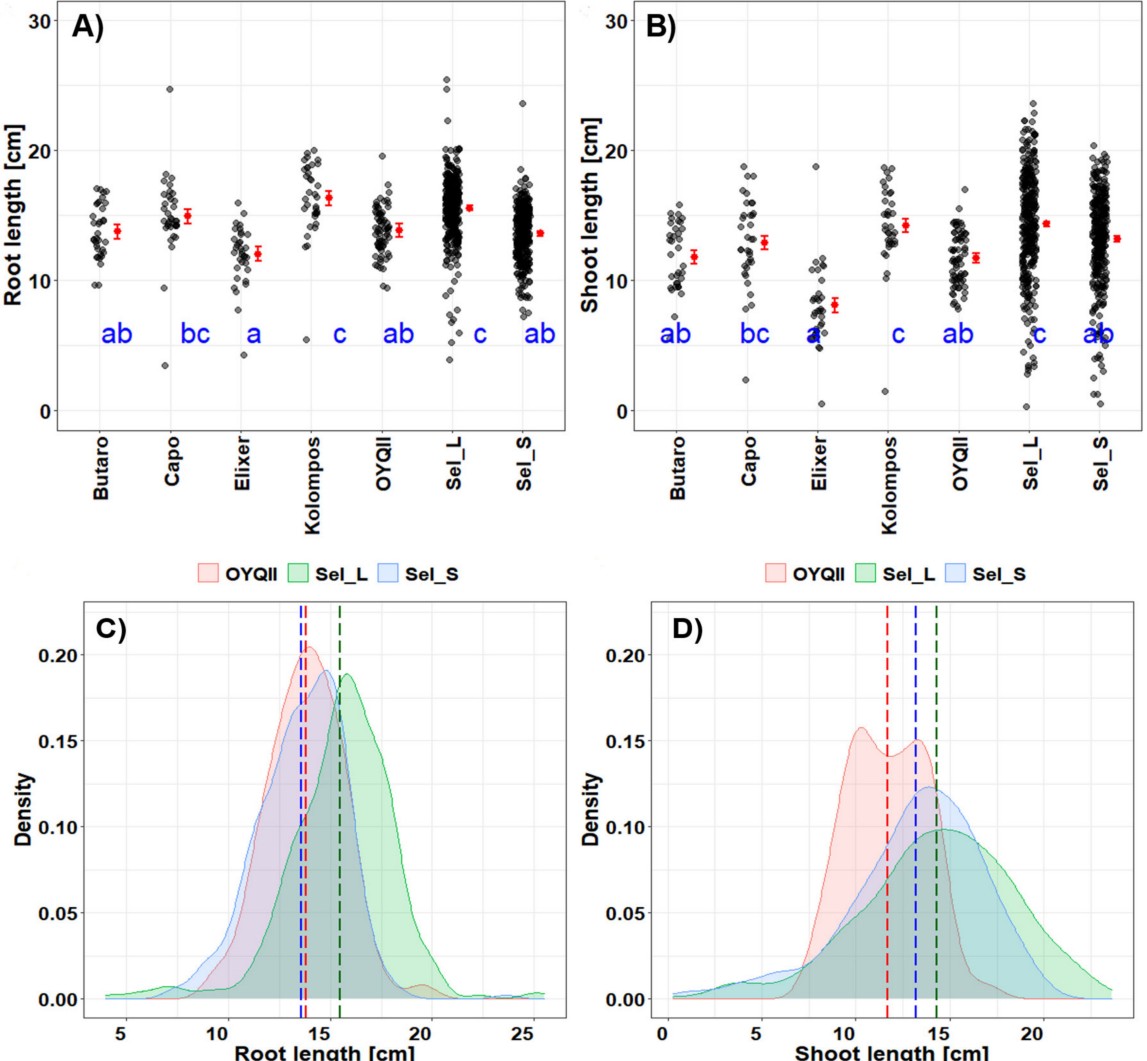

**Figure 2.** Seminal root length (**A**) and shoot length (**B**) of four reference varieties, the parental OYQII CCP and progenies of 9 lines each selected under hydroponic conditions either for long (Sel_L) or short (Sel_S) seminal root length after 14 days in the hydroponic system. Raw data are plotted (black dots) and estimated marginal means with standard error from linear mixed effect models (lmes, red). Different letters indicate significant differences at $p < 0.05$ estimated from lmes with pairwise comparison and Holm correction. Plots for individual progenies are shown in Figure S3. Density plots of seminal root length (**C**) and shoot length (**D**) for the parental Population OYQII the 9 Sel_L and Sel_S progenies. Dashed lines indicate means.

**Table 1.** Seminal root length (SRL) and shoot length (SL) increase relative to the mean of the parental OYQII CCP of nine progenies of plants selected for long (Sel_L) or short (Sel_S) seminal roots either tested under hydroponic conditions (SRL and SL) or in soil (SL only). Significant differences compared to the parental population are indicated with stars at $p < 0.05$ estimated from lmes with pairwise comparison and Holm correction.

| Progeny | Hydroponics Experiment | | Soil Experiment |
|---|---|---|---|
| | rel. Increase SRL (%) | rel. Increase SL (%) | rel. Increase SL (%) |
| Sel_L102 | 10.7 | 31.5 * | 38.3 * |
| Sel_L145 | 18.7 * | 13.5 | 30.1 * |
| Sel_L146 | 17.7 * | 23.2 | 26.1 |
| Sel_L191 | 17 * | 27.3 * | 34.5 * |
| Sel_L198 | 7.8 | 31.9 * | 30.8 * |
| Sel_L214 | 5.4 | −1.3 | −3.9 |
| Sel_L56 | 18.9 * | 31.3 * | 30.1 |
| Sel_L83 | 10.8 | 19.6 | 27.3 |
| Sel_L85 | 4.2 | 22.9 | 23.9 |
| Sel_S120 | 8.2 | 25.5 | 10.2 |
| Sel_S149 | −5.8 | 16 | 15.1 |
| Sel_S187 | 5.8 | 8.9 | 17 |
| Sel_S213 | −5.6 | 9.2 | 21.6 |
| Sel_S33 | −3.1 | 2.2 | 4.4 |
| Sel_S37 | −5.3 | 10.9 | 22.9 |
| Sel_S46 | −20.5 * | 27 * | 35.2 * |
| Sel_S57 | 1.3 | 4 | 6.5 |
| Sel_S58 | 10.8 | 7.9 | 25 |

As was the case for SRL, SL of Elixer was shortest (SL = 8.1 cm) and the Sel_L progenies were tallest (14.3 cm), significantly taller than the OYQII CCP (11.7 cm). Shoot length increase for the Sel_L progeny compared to OYQII CCP was 2.6 cm ($p < 0.05$), i.e., a relative increase of 22.2% with a clear population shift and for Sel_S 1.5 cm, i.e., 11.3% (not significant) (Figure 2B,D). Four of nine Sel_L progenies had significantly longer SL than the OYQII CCP. Additionally, one Sel_S grew taller than the OYQII CCP (Table 1, Figure S3).

Comparing all entries, neither the Sel_L nor Sel_S progenies differed significantly from OYQII for RFW or RDW. Elixer had the lowest (0.064 g) and Kolompos the highest (0.112 g) RFW, while Sel_S37 had the lowest (0.005 g) and Kolompos the highest (0.009 g) RDW. The selected progenies also did not differ in SFW or SDW from the OYQII CCP. Elixer had the lowest (0.088 g) and Sel_S57 the highest (0.177 g) SFW. Butaro had the lowest (0.011 g) and Sel_L187 the highest SDW (0.014 g) (data not shown).

In the soil evaluation experiment, the roots of the single plants were often entangled with the root sleeves, hampering the removal of the soil impeding root measurement. Consequently, there was no discernible genotype effect on SRL in the soil evaluation experiment of single plants (data not shown). In contrast, the effects on SL were statistically significant and followed the same pattern as in the hydroponic evaluation (Table 1, Figure 3) with a correlation of rs = 0.85 of SL between hydroponic and soil data ($p < 0.001$, Figure S4). The mean SL of the Sel_L progenies (39.8 cm) was significantly greater than that of the OYQII CCP (31.6 cm) and the Sel_S progenies (37.0 cm).

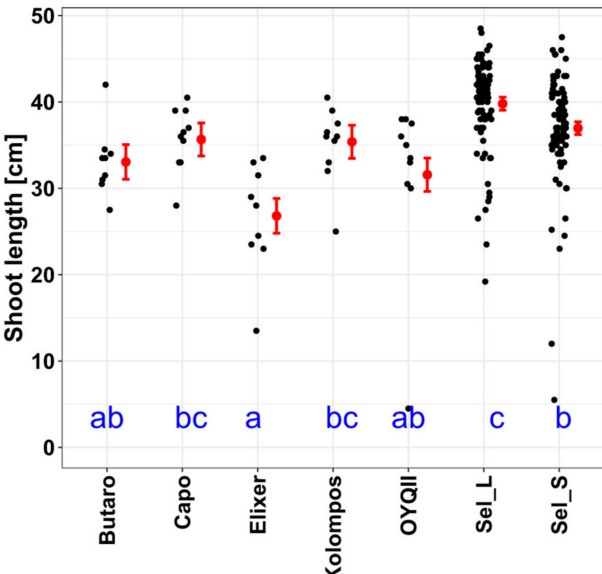

**Figure 3.** Shoot length of four reference varieties, the OYQII CCP, and progenies of nine lines each selected under hydroponic conditions either for long (Sel_L) or short (Sel_S) seminal root length when grown as single plants in pots with soil. Raw data are plotted (black dots) and estimated marginal means with standard error from linear mixed effect models (lme, red). Different letters indicate significant differences at $p < 0.05$ estimated from lmes with pairwise comparison and Holm correction.

### 3.2. Competition Experiment with Mustard

3.2.1. Soil Cover, Plant Length, Root and Shoot Biomass

In the competition experiment, mustard soil cover was not significantly reduced in the mixture with wheat, and there were no wheat-genotype-based effects on mustard cover (Figure S5B) nor root and shoot biomass (Figure S6A,B). Wheat cover significantly differed among wheat genotypes and was the lowest for Elixer (14%) and highest for Kolompos (27.5%). Wheat cover was significantly reduced in mixtures with mustard compared to monoculture with no differences between the Sel_S and Sel_L progenies and the OYQII CCP and no significant interactions (Figure S5A).

Elixer grew shortest (15.7 cm) and Sel_L56 (25.7 cm) tallest. Overall, mixing with mustard had no statistically significant effects on wheat plant length; however, plant length in Elixer increased significantly, while it was significantly decreased in Sel_L191 in mixture (Figure S7). As the plants had been grown for six weeks and pot depth was only 14.5 cm, only root weights could be determined.

Fresh and dry weights of wheat and mustard corresponded well, and only dry weight data are presented. Elixer had the lowest RDW of 0.6 g and Sel_S213 the highest of 1,1 g. There were no significant differences among genotypes for RDW, and only a tendency for biomass reduction in mixture with mustard (0.9 g) compared to pure wheat (1.0 g) with no differences between Sel_L and Sel_S and OYQII CCP (Figure 4A). In contrast, genotype and mixture effects were significant for SDW with a significant genotype–mixture interaction (Figure 4B). SDW was lowest for Elixer with 1.4 g and highest for Sel_S213 with 3.0 g. In two of the three Sel_L progenies and one of the three Sel_S progenies, SDW was not significantly affected by the presence of mustard, and the OYQII-CCP was negatively affected ($p < 0.05$).

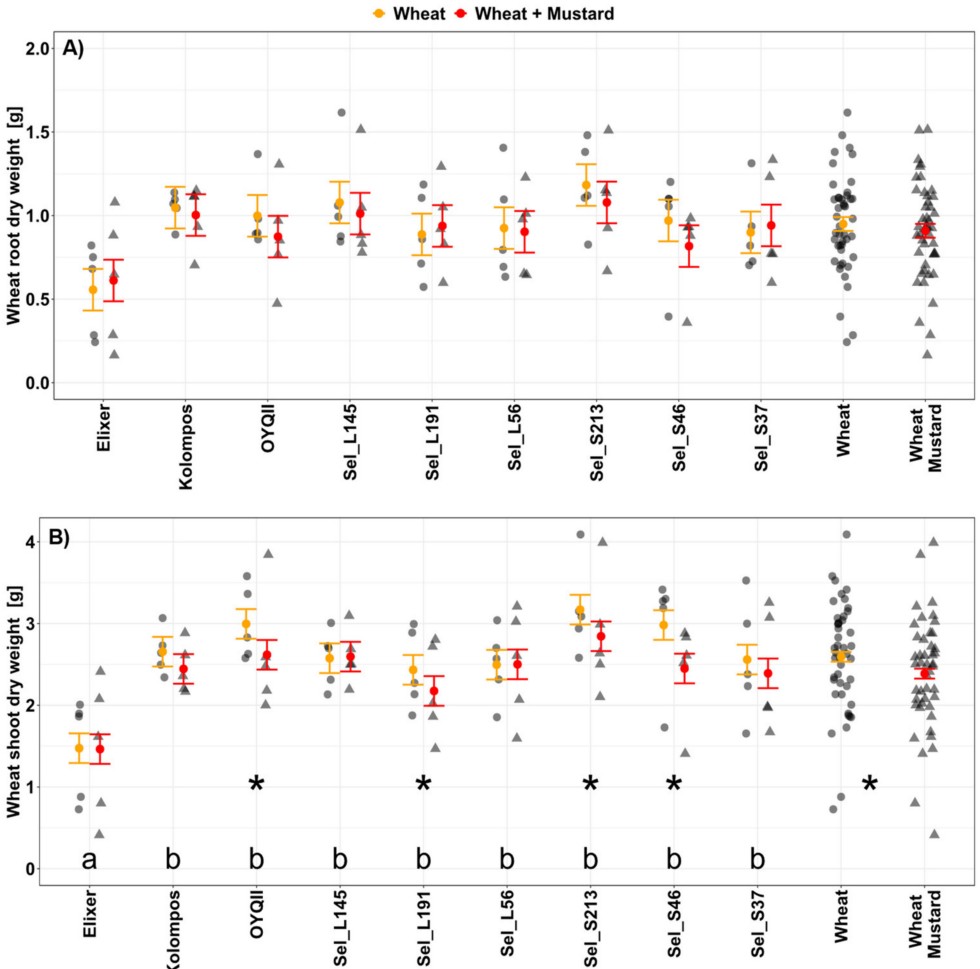

**Figure 4.** Root (**A**) and shoot (**B**) dry weights of six-week-old wheat plants grown either in pure stands or in the presence of mustard as a model weed. Raw data are plotted with dots for wheat monoculture and triangles for wheat in mixture with mustard, and overlapping data points are black. Estimated marginal means with standard error from linear mixed effect models are included for both pure wheat treatments (orange) and mixed wheat treatments (red). Different letters indicate significant differences between wheat genotypes across systems and stars between mixture and monoculture at $p < 0.05$ estimated from lmes with pairwise comparison and Holm correction. Sel_L are progenies of plants selected for long roots, and Sel_S are progenies of plants selected for short roots.

### 3.2.2. Wheat Leaf Traits

Elixer had the shortest leaves (12.0 cm) and Sel_L56 the longest (18.3 cm). While, overall, leaf length did not change significantly in the mixture compared to pure wheat, there was a significant genotype–system interaction. Leaf length of the Sel_L_56 progeny increased significantly from 17.3 cm in monoculture to 19.3 cm in mixture (this progeny was also not reduced in shoot biomass in mixture), while leaf length of the Sel_L_191 progeny decreased significantly from 16.2 cm in monoculture to 14.6 cm in mixture. Leaf length of Elixer also increased significantly from 11.3 cm in monoculture to 12.7 cm in mixture (Figure S8A). There was a significant genotype, but no system effect for leaf width, with the lowest leaf width found in Sel_S_46 (0.49 cm) and the greatest leaf width for OYQII (0.57 cm) (Figure S8B). Leaf area followed the same pattern as leaf length and tended to be reduced in mixtures. Elixer had the smallest leave area (4.5 cm$^2$) and Kolompos the largest (7.8 cm$^2$) (Figure S8C).

### 3.2.3. Competitive Response of Wheat

Cr for wheat cover varied between -44.4% (Elixer) and −23.1% (Sel_S37), for plant length between −4.7% (Sel_S46) and +6.7% (Elixer), and for leaf length between −10.2 % (Sel_L191) and +11.7% (Elixer). There were no obvious differences among the Sel_L and Sel_S progenies (Table 2). In contrast, Cr for RDW and SDW differentiated the two progeny groups with mean RDW and SDW reductions being greater for the Sel_S than for the Sel_L progenies. However, variation within the progeny types was very high (Table 2, Figure 5).

**Table 2.** Observed competitive response (CR) of wheat as relative change of wheat in mixture compared to wheat in monoculture calculated as (traitmix/traitmono)-1*100.

| Genotype | Cover | Plant Length | Leaf Length | RDW | SDW |
|---|---|---|---|---|---|
| Elixer | −44.4 * | 6,7 * | 11.7 * | 10.0 | −0.7 |
| Kolompos | −33.3 * | −2.0 | −2.6 | −4.2 | −7.9 |
| OYQII | −34.4 * | −3.0 | −3.2 | −12.5 | −12.6 * |
| Sel_L145 | −33.3 * | −0.9 | −3.4 | −6.2 | 0.8 |
| Sel_L191 | −38.5 * | −4.5 * | −10.2 * | 5.7 | −10.6 * |
| Sel_L56 | −29.2 * | 2.6 | 11.1 * | −2.4 | 0.2 |
| Mean Sel_L | −33.7 | −0.9 | −0.8 | −1.0 | −3.2 |
| Sel_S213 | −31.0 * | −0.6 | −4.1 | −8.8 | −10.3 * |
| Sel_S37 | −23.1 | 1.5 | 0.6 | 4.6 | −6.6 |
| Sel_S46 | −34.5 * | −4.7 | 0.4 | −15.8 | −17.8 * |
| Mean Sel_S | −29.5 | −1.3 | −1.0 | −6.7 | −11.6 |

* Stars indicate significant differences between mixture and monoculture at $p < 0.05$ estimated from two- factorial lmes with pairwise comparison and Holm correction. Statistical results are based on the model for absolute trait data.

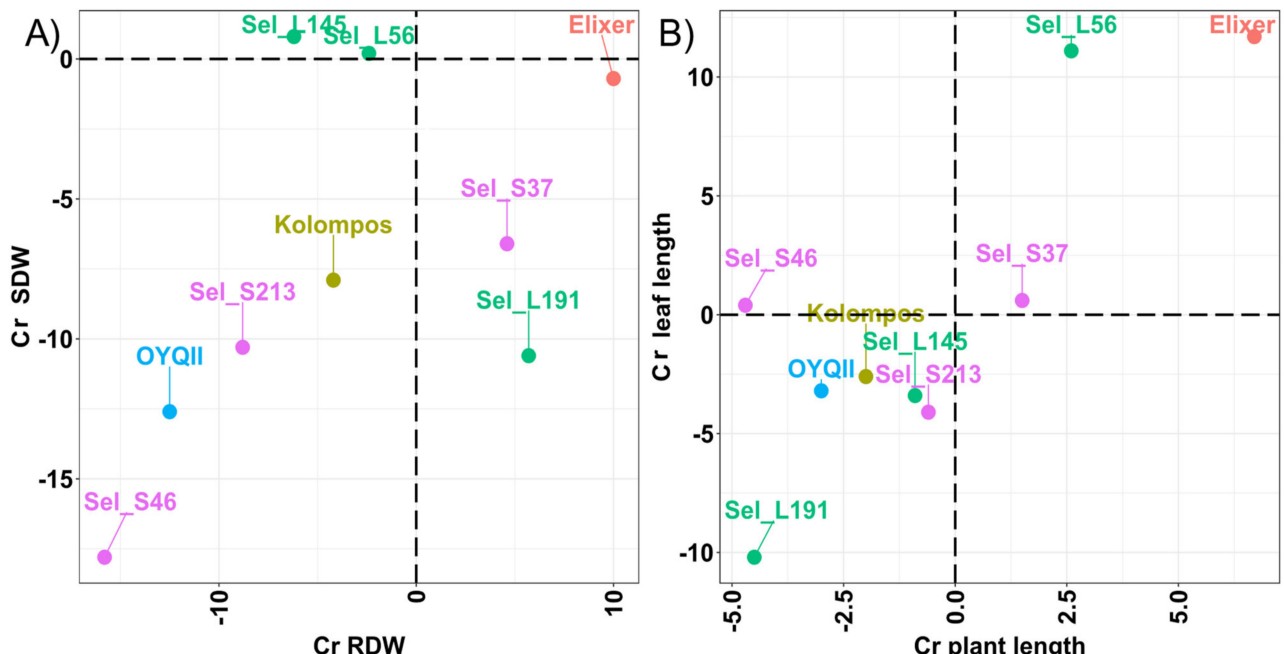

**Figure 5.** (**A**): Competitive response (Cr) of wheat for shoot dry weight (SDW) and root dry weight (RDW). (**B**): Cr of wheat for leaf length and plant length.

## 4. Discussion

### 4.1. Selection Effects on Root Vigor

The significant increase of seminal root length (SRL) in a single generation indicates that non-destructive selection under hydroponic conditions for a vigorous root phenotype is possible and feasible. This heritable trait change is a prerequisite for effective selection within a population but can also be useful for selection from crosses in early generations. A single round of selection for SRL under hydroponic conditions led to a statistically significant mean increase in SRL of 12%, while shoot length (SL) increased by 22%. In contrast, selecting for shorter SRL led to no change in SRL but an increase of 11% in SL. These data correlated well for SL when progenies were sown in soil, while data on SRL in soil could not be determined in this study, and seed needs to be increased before the experiments can be repeated with more genotypes and replicates to increase statistical power of the results.

Little published information on genetic gain for early vigor is available. The selection effect for SRL in a single generation achieved in our experiments is within the range achieved by Zhang et al. [38,39] for leaf width, leaf area, and biomass of 7.1%, 10.3%, and 5.3% per generation, respectively. Studies found that selecting with an electrical capacitance method for larger roots had a considerable effect, while selection for small roots had only a marginal effect in wheat [40] and barley [41]. Heřmanská et al. [40] estimated that selecting for larger roots resulted in a heritability of 0.43 and for smaller roots of 0.15, confirming our results of a heritability for SRL for Sel_L lines of 0.59 and for Sel_S lines of 0.21. Natural selection of the YQ CCP (and other CCPs) from the F6 to the F11 under organic conditions resulted in significantly longer seminal roots compared to natural selection under conventional conditions [23], and SRL in hydroponic conditions correlated well with early soil cover in the field [24]. However, overall changes in SRL were rather small over the 10 generations [23]. In contrast to the natural selection and adaptation in the field, the genetic gain and heritability for SRL achieved with non-destructive selection here is substantial and relevant for practical plant breeding. The significant but moderate correlation of SRL with SL suggests either a moderate degree of linkage of loci for both traits or pleiotropy. This indicates sufficient variation to select for different trait combinations such as high SRL and high SL or high SRL and low SL. If both early vigor in roots and long shoots are the target, then a simple strategy could be to select for increased seedling shoot vigor and indirectly for root vigor.

Compared to German cultivars, Kolompos is an early maturing cultivar due to its Hungarian origin, and it performed well in dry springs in Germany [27]. Early plant development and maturity is a strategy in plants to escape summer drought and is used in wheat breeding for dry environments [42]. Therefore, screening for early vigor in hydroponics could be an efficient method to screen for genetic material that confers advantages under drought conditions, provided early vigor reliably results in earlier plant development.

Multiple traits contribute to weed suppression with key traits including early vigor, final plant length, leaf area, and biomass [43]. In principle, it should be possible to combine all traits contributing to weed suppression in a single cultivar. Combining selection of early vigor of roots with early vigor of shoots might increase genetic gain for early vigor in general. Improving SRL has a great potential since a range of studies linked fast early root growth to improved uptake of nitrogen [15,16], phosphate, and magnesium [44], and it also positively impacts grain yield in the field [45]. While plant length usually reduces harvest index in genetically homogeneous materials and therefore grain yield (trade-off effect), we have observed that harvest index and yield do not necessarily follow the same trajectory in CCPs (Baresel, Finckh et al., submitted). Early vigor in itself does not necessarily reduce harvest index.

The parental OYQII CCP seeds had been in storage for two years before use in the hydroponic comparison with their selected progenies. This is within the five-year storage period that did not affect performance of seedlings under hydroponic conditions [24]. Thousand-grain weight (TGW) did not play a role in the comparison between the parental

population and the progenies, and despite the significantly lower TGW in the Sel_L progenies, measured traits were similar to Kolompos. Thus, we are confident that the results reflect true selection effects and therefore provide a proof of concept that non-destructive hydroponic root selection for early vigor can be applied. All selected progenies that produced seed in the greenhouse are currently being multiplied in the field for further testing as lines and as experimental populations. Conclusions with respect to the genetic basis of this observed phenotype shift will require genetic screening, for example, of known QTLs for seedling root vigor in wheat as described by Atkinson et al. [45].

### 4.2. Competetive Response of Wheat and Competetive Effects on Mustard

In our experiment, while wheat cover was reduced in the mixture, mustard cover was not, indicating that the competitive effects on mustard were negligible. White mustard is a strong competitor for wheat compared to other common weeds such as *Lolium multiflorum*, *Chenopodium album*, or *Stellaria media* due to its fast emergence and early growth [46], and higher soil nitrogen content has been shown to increase the competitive effect of mustard on wheat [47]. Nitrogen levels in our experiment were also extremely high (an equivalent of 339 kg ha$^{-1}$) and as such may help to explain the results. Although wheat grew taller than mustard, this occurred only relatively late. Thus, mustard largely overgrew the wheat early on, and early light interference of mustard on wheat resulted in a strong negative competitive response of wheat with respect to soil cover. The lack of significant competitive effects of wheat on mustard and also no specific wheat genotype dependent competitive effects on mustard was thus likely due to asymmetric competition in favor of mustard. Abdolahi et al. [48] showed that wheat genotypes vary in their competitive effect on mustard biomass, resulting in mustard biomass reduction between 14.4% and 67.4% with some genotypes showing superior effects, although no traits were mentioned conferring these advantages. Despite these suboptimal conditions, differences in competitive response for certain traits became evident in our experiment. Thus, despite the fact that Elixer grew taller with increased plant length (Figure S7) and leaf size (Figure S8A), it still had the greatest reduction in soil cover when grown with mustard (Table 2). Absolute values for RDW, SDW (Figure 4), plant length (Figure S7), leaf length, and leaf area(Figure S8B,C) were smallest for Elixer, characterizing it as a conventional ideotype. This is in line with field results where Elixer showed similarly poor early soil cover compared to the YQ CCP [24], and it was considerably less suppressive to weeds on-farm [6].

The competitive response of wheat in general was small and highly variable. Only for SDW, the Sel_L lines seem to have a slightly reduced Cr compared to the Sel_S lines. The overall observed effect of selection on the competitiveness of wheat progenies was weak. In part, this is likely due to the low numbers of progenies studied, the high nutrient levels, and the fact that only one round of selection was carried out. However, the behavior of individual wheat progenies delivers some interesting insights. Sel_L56 increases its leaf and plant length in mixture with mustard similar to Elixer but in contrast to other progenies. This could be due to a variation in shade avoidance strategies that was demonstrated previously for wheat with consequent yield impacts [49]. Shade avoidance allows plants to detect competitors by a changed red/far-red ratio due to surrounding competitors and stimulates growth in plants to avoid shading [50]. Shade avoidance can be maladaptive in wheat monocultures in terms of population yield [51] but adaptive in terms of community yield if the cropping system is diversified and plasticity aids optimal niche complementarity, e.g., in grassland species mixtures [52]. Hypothetically, this could hold true for crop species mixtures [53]. Consequently, genotypes such as Sel_L56 that might have increased shade avoidance and show high absolute early vigor in SL, SRL, leaf, and plant length might be promising in diversified cropping systems. The lack of changes in leaf width of all cultivars and progenies confirms a very high narrow sense heritability of this trait [54] and hence low plasticity in response to environmental factors.

### 4.3. Outlook for Future Breeding

Breeding approaches that combine efficient selection methods for genetic gain and maintain intraspecific diversity for stress resilience of crops are highly needed. Results from the hydroponic experiments demonstrate the potential of this system for non-destructive selection of root vigor, allowing for a substantial genetic gain for SRL on a population level, provided a high enough number of seedlings are selected. To enhance the effects, additional rounds of selection could potentially be useful to increase genetic gain to a magnitude that results in practically relevant increases in competitive ability. For this purpose, an additional selection tool for roots is selection by means of electrical capacitance, as this allows non-destructive field sampling [55]. However, this method also has limitations such as the influence of soil moisture on electrical capacitance. Modern image-based methods to measure early vigor in the field could also be promising [56]. Such selection methods could be combined to inform the selection process of genetic materials for crossings, selected for both early vigor of roots and shoots. An interesting approach could be to combine hydroponic-based root selection with field- based evaluations of reassembled populations, as well as admixtures to existing populations.

Long-term breeding strategies would need to monitor and manage both genetic gain and intraspecific diversity. As selection decreases diversity, which would be the case if only selected lines are reassembled into a population, multiple populations could be maintained that are selected for a certain trait similar to the approach of group selection suggested by Wuest et al. [57].

The genetic material evaluated in this study together with the additional lines that are to be evaluated are a starting point for further breeding programs. Thus, the best Sel_L progenies for SRL and SL could be crossed with chosen pure lines and with progenies of the original population with additional traits of interest for improved competitive response to enrich alleles for early root vigor. For example, Sel_L56 seems to combine some interesting traits, including a high genetic gain for SRL (18.9%) and SL (31.3%) compared to OYQII (in hydroponics), increase of leaf length, and area in mixture with mustard (shade avoidance). Sel_L56 is also the genotype with the greatest plant length and (apart from Kolompos) the largest leaf area. Kolompos is also promising in terms of yield and land use efficiency in species mixtures under dry conditions [27].

To confirm changes in competitive response and competitive effects, additional experiments are needed under more realistic N-levels including a higher number of progenies and also multiple weed species or additional crop species suitable for a cereal mixed cropping system.

**Supplementary Materials:** The following are available online at https://www.mdpi.com/article/10.3390/su132212778/s1, Figure S1 Correlations for traits individual plants, Figure S2: Correlations for traits averaged, Figure S3: Models for root length, shoot length for individual progenies, Figure S4: Correlations between traits in hydroponics and soil, Figure S5: Soil cover for wheat and mustard, Figure S6: Mustard weights, Figure S7: Length of wheat plants, Figure S8: Length, width and area of wheat leaves, Table S1: List of statistical models.

**Author Contributions:** Conceptualization, M.R.F. and J.T.; methodology, J.T. and M.R.F.; validation, J.T., M.R.F. and O.D.W.; formal analysis, J.T.; investigation, J.T.; resources, M.R.F.; data curation, J.T.; writing—original draft preparation, J.T.; writing—review and editing, J.T., M.R.F. and O.D.W.; visualization, J.T.; supervision, M.R.F. and O.D.W.; project administration, M.R.F.; funding acquisition, M.R.F. All authors have read and agreed to the published version of the manuscript.

**Funding:** This research was a part of the project ReMIX "Redesigning European cropping systems based on species MIXtures" funded by the EU's Horizon 2020 Research and Innovation Programme (Grant Agreement No. 727217).

**Institutional Review Board Statement:** Not applicable.

**Informed Consent Statement:** Not applicable.

**Acknowledgments:** We thank Matthias von Ahn for his careful support in the selection and competition experiment and Georg Saathoff, Sven Heinrich, and Rainer Wedemeyer for their strong support of the competition experiment. Elsa Zwicker was crucial in the laboratory to prepare the high-quality nutrient solutions. The whole research group at the Department of Ecological Plant Protection provided an excellent and supportive research environment.

**Conflicts of Interest:** The authors declare no conflict of interest.

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
