# Peer review of "Combining Genetic Gain and Diversity in Plant Breeding: Heritability of Root Selection in Wheat Populations"

_sustainability, doi:10.3390/su132212778_

Round 1
Reviewer 1 Report
Dear Authors
In the attached file you will find my comments on the manuscript coded as 'sustainability_1428742'.

Author Response
Dear Reviewer 1,
thank you very much for your review, improving our research substantially. Attached you find a file adressing all your comments and suggestions.
Best wishes, Johannes Timaeus

Reviewer 2 Report
According to the definition, a CCP is a group of plants that results from a heterogeneous parental source, which can be reproduced without undergoing modifications after having adapted to the pedo-climatic conditions, specific to a region and is obtained by crossing diallel of at least five genotypes, or from a different crossing protocol, but which leads to obtaining an equally diverse population. Crossbreeding of genotypes is followed by the cultivation of offspring and their exposure to natural selection and supervised selection by breeding in successive generations.
In the present study were tested if a hydroponic system can be used to non-destructively select root traits from CCPs for high early vigor in order to increase genetic gain, while maintaining population diversity. More specifically, the authors try to respond to the questions if is selection for seminal root length in hydroponics heritable and how strong is the genetic gain achieved in one breeding cycle on population level?
To answer these questions 40 wheat progenies were selected from a CCP for high seminal root length (SRL) and 40 for short SRL in hydroponics and grown to maturity in soil substrate.Plants progenies were evaluated for selection effects in a hydroponic system. In addition, progenies with sufficient seed quantities available, were additionally tested in soil substrate. Finally, the parental CCP and selected progenies were evaluated for their competitive effect compared with the model weed-mustard.
Therefore, the manuscript is well documented and drawn up, with an updated introductory part, the material and methods used are modern based on a careful choice and handling of biological material, hydroponic method of selection for the character of the length of the roots, evaluation and propagation of seeds and experiments on competitiveness testing.
Also, data collection, processing and statistical interpretation of it provide valuable scientific results that demonstrate the potential of this system, in the non-destructive evaluation and selection for the characters regarding the vigor of roots at the population level. Obviously, if the studies can be extended to other cultivated species, assessing other types of characters and inducing competitiveness towards more weed species, the future research will be able to prove this fact.
A very good manuscript.
Congratulations!
Author Response
Dear Reviewer,
thanks a lot for your fast review and the very positive evaluation. This pushes us and our motivation that our research is good and relevant. We will improve the English by a native speaking colleague.
Best wishes, Johannes Timaeus
Reviewer 3 Report
The authors tried to combine genetic gain and diversity in wheat breeding using non-destructive selection for root vigor from wheat populations. I just have a major comment. I suppose hydroponics system for non-destructive selection is just a tool. Therefore, it doesn't need to be emphasized in the title and main question. My minor comments are in the attached file. Thanks

Author Response
Dear Reviewer,
thank you very much for your review, improving our research substantially.
In the attached document you find a document adressing all your suggestions and comments.
Best wishes, Johannes Timaeus

Round 2
Reviewer 3 Report
I don't see any issues in the manuscript.
I suppose the manuscript is publishable in Sustainability.
Thanks,